# Exploring Individual Components of Sport Persistence in the Light of Gender, Education, and Level and Type of Sport

**DOI:** 10.3390/jfmk9040205

**Published:** 2024-10-25

**Authors:** Benedek Tibor Tóth, Hanna Léna Tóth, Csanád Lukácsi, Oszkár Csaba Kocsner, Buda Lovas, Bence Tamás Selejó Joó, Hanna Czipa, Regina Bódi, Zsuzsa Lupócz, Rozália Paronai, Mátyás Kovács, Karolina Eszter Kovács

**Affiliations:** 1Institute of Psychology, University of Debrecen, 4032 Debrecen, Hungary; bene.toth@gmail.com (B.T.T.); hannalenatoth@gmail.com (H.L.T.); lukacsi.csanad.pal@gmail.com (C.L.); kocsner.oszi@gmail.com (O.C.K.); benui1999@gmail.com (B.T.S.J.); czipahanna@gmail.com (H.C.); bodiregina99@gmail.com (R.B.); zsuzsalupocz@gmail.com (Z.L.); prozka2002@gmail.com (R.P.); 2Institute of Psychology, Eötvös Loránd University, 1064 Budapest, Hungary; lovasbuda@gmail.com; 3Faculty of Chemical Technology and Biotechnology, Budapest University of Technology and Economics, 1111 Budapest, Hungary; matyikaa2000@gmail.com

**Keywords:** sport persistence, ecological model, qualitative research, individual motivation, grounded theory

## Abstract

**Background/Objectives:** Sport persistence can be coded as an indicator of sport performance and commitment, incorporating personality traits such as resilience, adaptive coping, and positive personality traits. Thus, athletes do not merely persist in sporting activities but are qualitatively committed to them. **Methods:** In the present research, we used a qualitative methodology to investigate the factors underlying sport persistence, using Bronfenbrenner’s socio-ecological model. In total, 133 high school and college student-athletes were surveyed in a semi-structured interview study. We set the following research question: How do the factors involved in developing sport persistence vary across athletes’ gender, level of study, and level and type of sport? Data were analyzed along the lines of grounded theory. ATLAS.ti and IBM SPSS 22.0 statistical software were used for the analysis. **Results:** Our analysis divided individual motivation into intrinsic (health promotion, habituation, becoming a competitive athlete, self-improvement, self-actualization, relaxation, and sport enjoyment) and extrinsic motivation (family-related motivation, coach, social relationships, competition, livelihood, recognition). The cross-tabulation analysis revealed that contrary to the general trend, no significant differences can be experienced in the core motivational patterns contributing to sport persistence. However, significant differences could be detected concerning the level of education, level of sporting activity, and type of sport. **Conclusions:** Our findings not only shed light on the unique factors underlying sport persistence, but also challenge the trends observed in traditional sport motivation analyses. This insight could potentially revolutionize how we approach youth sport promotion and physical activity among young people, making our research highly relevant and impactful.

## 1. Introduction

Sport persistence can be considered as the pinnacle of sports performance and commitment. It incorporates personality traits such as resilience, adaptive coping, and positive personality characteristics. Thus, athletes persist in sports activities and are qualitatively committed to them [1,2]. This behavior and effectiveness are encompassed by sport persistence, which is not widespread in international practice, as research typically focuses on sports habits, motivation, and commitment. Furthermore, existing research often views sport persistence as a binary variable, meaning that an athlete is considered persistent if they are still engaged in sports at the time of the study and have not dropped out. However, it is essential to capture persistence more accurately by creating a complex sport persistence indicator [1].

To understand the nature of sport persistence, one should start by discussing the most basic definitions related to it. Over recent decades, the definition of sport has undergone a considerable transformation. The variety of definitions associated with the concept of sport arises from its complex nature, shaped by cultural, historical, and societal influences, along with differing viewpoints regarding the aims, framework, and importance of physical activities. According to the European Charter of Sport [3], *sport* is defined as “any physical activity, whether performed occasionally or in an organized form, the purpose of which is to develop or improve physical and mental fitness, to create social contact or to achieve results in competition at various levels”. In contrast, the term exercise is more specific, characterized as “a specific type of physical activity that is planned, structured and repeatedly done to improve or maintain physical fitness”, while physical activity is broadly defined as “any movement produced by skeletal muscles that results in energy expenditure” [4], p. 128. According to the WHO [5], *physical activity* is any body movement generated by skeletal muscles that necessitates energy expenditure. This encompasses all forms of movement during leisure, transportation to and from locations, or as part of an individual’s occupational duties. Both moderate and vigorous-intensity physical activities contribute positively to health. The initiation of physical activity at a young age is vital, underscoring its significant role in overall development.

Motivation is an important background factor in the pursuit of a sporting activity. The reasons for engaging in a particular activity can be many and varied. Motivation can be classified into two broad categories, known as intrinsic and extrinsic motivation [6,7]. In the case of intrinsic motivation, the athlete plays sport for the sake of the activity itself, i.e., the activity itself acts as a rewarding force for him. This is the strongest and most persistent behavior, and the performance of the action itself acts as a positive reinforcer. Intrinsically motivated individuals formulate goals regarding competence, self-determination, excellence, and success [8]. In the case of extrinsic motivation, however, the activity is not intrinsic but is only the result of some external factor. Its source can be praise or discipline, reward or punishment (more challenging training, withdrawal of previous reward). It is typically a short-term and less effective type of motivation, as the removal of the external drive can significantly decrease the athlete’s performance [9]. The impact of external reinforcement on athletes is twofold. Rewards can increase arousal for the player. These factors can all have the opposite effect to that expected. Therefore, the player’s personality, activity level, and environment should always be considered. It is advisable to base rewards primarily on the game and the pleasure of playing, as well as the interest and need to win [10,11].

Based on research results related to sports habits, it is worth dealing with the issue of persistence in both domestic and international contexts and at various levels of sports. In elite and competitive sports, the role of persistence is unquestionable, as a successful sports career can only be achieved if the athlete remains committed to their activities, ideally with professional training, individually and guided by a coach, and ideally with psychological preparation [12,13,14,15]. Preventing dropout is essential because it takes time for the seeds of hard work to blossom into success. However, the significant physical, emotional, and mental changes occurring during adolescence can significantly complicate the process, often causing temporary setbacks in performance [16].

At the same time, it is worth examining sport persistence, not only in competitive sports. Recreational sports often lack the competitive component and are less effective in performance, as there are no tangible performance indicators, such as medals or recognitions. However, the role of persistent and enduring recreational sports is essential for personality development, especially during sensitive periods such as adolescence or young adulthood, as the effects of persistent sports are also detectable in recreational sports, albeit with a different focus [17].

Several factors influence sport participation. Gender differences in sports participation are shaped by biological, social, and cultural factors, with men generally having higher muscle mass and testosterone levels, impacting strength and speed [18,19,20,21]. Men tend to be motivated by competition and performance, while women are more influenced by social factors like team spirit and cooperation [22]. These differences are also shaped by traditional gender roles and media portrayals, making sports motivation a complex interplay of biological, psychological, and sociocultural influences [23,24].

Sports participation varies across education levels due to factors such as age, academic load, and available infrastructure [25,26]. In primary education, sports are often integrated into the curriculum, with students participating enthusiastically in physical activities that promote development and social skills [27]. In secondary education, interest in sports fluctuates, with daily physical education in some countries like Hungary, while academic demands limit participation for some students [28,29,30,31]. In higher education, although sports facilities are available, students often engage less due to academic pressures, although physical activity is encouraged for its positive effects on mental health and academic performance [16].

The examination of sports types is key to understanding youth sports programs’ effectiveness, with distinct psychological impacts between team and individual sports. Team sports promote social skills, teamwork, and community support, enhancing social capital, while individual sports focus on self-discipline and personal development but may lead to higher anxiety and isolation [32,33,34,35,36,37]. The choice of sports is also influenced by family and cultural backgrounds, affecting athletes’ cognitive and emotional growth [38,39]. Additionally, the intensity and frequency of participation affect sports outcomes. Beginner athletes typically engage in lower-intensity activities, while advanced athletes participate in high-intensity training for optimal performance [16,40,41]. Competitive athletes are highly committed, treating sports as a lifestyle with rigorous training, whereas recreational athletes focus more on health, stress relief, and social interaction with more flexible commitment [42,43].

In the present research, we focus on the foundations of sport persistence, formulating the following research question: What factors dominate the development of sport persistence among adolescent (high school) and young adult (university) athletes along Bronfenbrenner’s dimensions? In line with the research question, we categorize the participants’ responses into content categories and then perform comparisons based on several demographic and sport-specific variables, following this hypothesis:

**H1:** *The role of individual-level motivational components is significant in developing sport persistence. However, the weight of these factors varies by gender, sport type, level of sports participation, and educational career stage*.

## 2. Material and Methods

### 2.1. Sample

During data collection, we applied theoretical sampling. Initially, we identified a broader range of potential interviewees based on expert criteria, then selected further participants from this pool based on emerging concepts during the simultaneous analysis of the interviews. The sample was not random but guided by expert criteria and emerging concepts from ongoing data analysis, which allowed us to focus on relevant categories (e.g., gender, type of sport, and educational level) and ensure diverse representation, supporting the control of confounding and interactive factors. The categories revealed during open, axial, and selective coding guided the sampling. The expert criteria for determining the broader range of potential interviewees were as follows:Engagement in regular (at least three times a week) sports activities (any individual or team sport was suitable);Being engaged in secondary or higher education studies;Aged between 14 and 25 years.

The research was carried out between 2023 October and 2024 February. We used *semi-structured interview techniques* to conduct qualitative research with high school and university athletes. During recruitment, we contacted sports clubs and sports schools, where coaches and institution leaders were asked to support our research by letting the interviews be carried out in their institutions. Besides this official type of recruitment, online recruitment was also revealed to reach out to students studying at sports schools and university student-athletes. The research was carried out with the involvement of members of the Sport Persistence Research Group. Data were handled in accordance with GDPR and only members of the research team had access to the database. Responses were voluntary and anonymous, no personal data were collected, therefore, re-identification based on the answers given was not possible.

Overall, the mean age of the sample was 18.9 years (SD = 3.4), more specifically, 22 years in the tertiary sub-sample (SD = 1.9) and 17.15 in the secondary sub-sample (SD = 2.8). Sample characteristics are introduced in Table 1. 

### 2.2. Methods

#### Data Collection

In the research, data collection was carried out through *semi-structured interviews* while we aimed to have an in-depth analysis of the factors influencing the manifestation of sports persistence. The current paper focuses on the individual factors standing in the background of persistent sporting behavior. However, we should note that this is a slice of bigger research, focusing on the factors influencing sport persistence considering the various levels of the socio-ecological model. The interview schedule included four blocks, including sociodemographic background, sport-specific questions, competitive athletes, and sport persistence issues along the levels of the ecological model which block could be divided into five sub-sections (see Table 2). The interview structure and the questions were designed based on our previous analysis of the existing literature focusing on sport persistence [44,45].

### 2.3. Theoretical Background of the Analyzing Method

We analyzed the data using grounded theory (GT), an inductive and iterative method. With the help of grounded theory [46,47], we aimed to distance ourselves from our knowledge of the subject, continuously formulating theoretical explanations by progressively abstracting from the data obtained from the interview analyses. The semi-structured interview questions served only as a starting point, allowing unrestricted associative reflection to discuss issues relevant to the interviewees and the topic [48,49]. At the same time, we applied deductive category analysis to structure the analysis. We followed a qualitative, category-driven text interpretation, highlighting the possibilities of feedback and intersubjective testing. We used ATLAS.ti software (22.2.5 Student version) for content analysis. Following the methodology of GT, we followed a constructivist meaning-making process, focusing on the “what” and “how” questions to uncover the components shaping the sport persistence of high school and university athletes. We used GT techniques to identify themes and patterns and create categories of factors involved in developing sport persistence.

Grounded theory establishes the importance of highlighting contextual influences, e.g., intrapersonal characteristics (ontosystem), interpersonal relationships (microsystem), community influences (mesosystem), and cultural values and national characteristics (macrosystem), which can be used to reveal socio-psychological influences [47]. Taking this into account, we explored and thematically sequenced the characteristics of micro and macro environmental influences in the interview transcripts. This iterative process included both inductive and deductive reasoning. We applied inductive, deductive, and abductive reasoning during the interview analyses. The coding process was segmented into four distinct phases, adhering to the constructivist approach of grounded theory (GTC) [50]. The initial phase involved an in-depth familiarization with the interview texts through several readings and auditory reviews, succeeded by semantic-based open coding executed in the second phase. In the third stage, we categorized the samples of codes into distinct categories [51]. Subsequently, in the fourth stage, we engaged in iterative analysis cycles to scrutinize and enhance the code classifications within these categories. Throughout the analysis, we adhered to the principles of theoretical and personal triangulation, examining the same research question and the family–school relationship phenomenon through various theoretical frameworks and perspectives. This methodology enabled us to investigate different facets of the phenomenon by juxtaposing various theories, ultimately allowing for a more holistic understanding informed by diverse viewpoints, as noted by Flick [52]. Consequently, this study employed the quality criteria articulated by Flick [52], which include credibility, data-rich elicitation, commitment to fieldwork, and confirmability. By following inductive, deductive, and abductive reasoning during the coding process, we ensured that multiple factors were systematically considered and categorized, minimizing the effect of confounding variables.

#### Phases of Analysis

The type of qualitative study and the use of grounded theory indicate moving in flexible environments, in terms of facing unpredictable fields of action. For this reason, the research was structured in six phases, including the following tasks carried out by the research group members:Phase 1: definition of the sample (the head of the research group, based on the research project and its plan, detected the potential sample);Phase 2: validation of the interview script (all members were asked to think through the potential interview questions, then, a discussion was carried out in a research group meeting);Phase 3: application of the interviews (all members were involved in the interview process, including recruitment and application as well);Phase 4: transcription and analysis (five research group members (HC, RP, HLT, OCsK, CsL) were involved in the preparation of the transcripts and seven in the analysis (BTSJ, RB, ZsL, BTT, BL, MK, KEK));Phase 5: coding and categorization (seven five research group members (BTSJ, RB, ZsL, BTT, BL, MK, KEK) were involved in the creation of codes. The initial stage of the analysis involved the identification and coding of incidents. Afterward, the preliminary codes were compared with other codes, and the codes were grouped into categories. The incidents within each category were then compared with those in other categories. Then, the future codes and categories were compared. Subsequently, the new data were subjected to comparison with the information that had been collected at an earlier stage of the analysis);Phase 6: interpretation (all members were asked to support the interpretation of the results).

### 2.4. Statistical Analysis

For content analysis, we used ATLAS.ti software (22.2.5 Student version). The analysis of content was performed utilizing the ATLAS.ti software (22.2.5 Student version); however, its application was limited to the synthesis of coding. In this examination, data were organized around factors at the microsystem level, which emerged from a combination of insights gained from the interviews and relevant concepts derived from existing literature. The ATLAS.ti software demonstrated significant utility in managing qualitative data and investigating the interconnections among complex concepts. It enabled precise management of the coding process and the organization of themes and sub-themes, thereby aiding in the development of typologically arranged categories.

Beyond content analysis, we examined the differences in the distribution of various sport motivational components and micro-, meso-, and macro-system factors influencing sport persistence based on gender, type of sport, level of sports participation, and study level using IBM SPSS (version 22.0) statistical software. Due to the nature of the data (categorical variables), cross-tabulation analyses, and chi-square and Fisher tests were carried out. Statistical significance was considered as *p* < 0.05 (two-tailed).

## 3. Results

### 3.1. Individual Motivational Components Behind Sport Persistence

First, we examined the individual motivational components behind sport persistence. Based on the content analysis, we established two significant groups: external and internal motivational components. The internal motivation factors included health maintenance, habit, becoming an elite athlete, self-development, self-actualization, relaxation, and love of sports. In contrast, the external motivation group included family motivation, coaching, social relationships, competition, livelihood, and recognition. The following sections describe the subcategory characteristics of the two main motivational factors. Table A1 (Appendix A) introduces some examples concerning each category.

As **internal motivational components**, we grouped seven factors based on the respondents’ answers. Regarding health maintenance as a motivational component, interviewees highlighted that sports help maintain fitness and a healthy lifestyle. A good physique and physical well-being are also important aspects. Additionally, sports are seen as an activity that provides mental relaxation, contributes to stress relief, and thus helps maintain health. For some, sports like horseback riding represent freedom and social bonding. Respondents highlighted sports’ various physical and mental benefits, including better well-being and building social relationships.

The habit as a motivational component included responses reflecting that sports are an integral part of the participants’ lives, often a daily activity since childhood that gives them strength and completes their lives. Training and physical activity are indispensable for them and deeply embedded in their daily routines. The becoming an elite athlete component included responses where participants were mainly motivated to become recognized and high-level athletes. They all aim to play professionally and achieve outstanding performance in sports, advancing their careers and gaining recognition. In the self-development motivational component, the focus is on future-oriented personal and physical development. Respondents highlighted the importance of belonging to a community and physical and mental growth. Many are motivated by achievements and continuous challenges. Some aim to reach a professional level, become recognized athletes, or build a successful career. Sports help them focus, persevere, and constantly improve. Talent exploitation, passion for sports, and family influences also play significant roles.

Self-actualization, a specific outcome of self-development, also appeared as a declared motivational factor. Some respondents enjoy sports because they are good at them and pursue them to realize their dreams, giving them additional strength and purpose. Relaxation, as a motivational segment, focuses on sports’ relaxing, stress-relieving, and mentally refreshing effects. Sports help them relax, forget problems, make them happy, and energize them. Many mention that exercising feels good, is entertaining, and contributes to a healthy lifestyle. The community experience, company, and hobby aspects also play essential roles. Overall, sports provide significant mental and physical regeneration for the participants. Finally, the love of sports constitutes the last internal motivational factor, indicating that the primary motivation for the participants is their love and enjoyment of sports. For many, the most important factors are the love of sports, joy of movement, hobby, and passion. Many mention that sports are a part of their lives, energize them, and are missed if skipped. Sports’ community and health maintenance roles are also important, along with love for the team and coach as motivating factors. Overall, participants engage in sports because they love it and it brings them joy.

As **external motivational factors**, we identified six categories. Family motivation refers to the participants’ sports activities primarily initiated by family traditions. One participant started sports because they studied it at university, and their father also practiced it, making it a family tradition. Another participant started because of their family, which has become an indispensable part of their life. The coach also emerged as an external motivational factor. The participants’ sports activities were significantly motivated by respect and appreciation for their coach. They mentioned their coach’s excellence, and the coach’s encouragement, support, and positive reinforcement were also emphasized. Social relationships as external motivational components were also significant.

Additionally, love for the company and team, the sport, and the enjoyment of time spent with teammates were influential. Passion for competitions, training, and development also play an essential role. Sports represent a communal experience, team spirit, and joy for them. Competition emerged as a distinct aspect. This component centers on the love of competition, individual or team athlete. Competing against each other contributes to personal development and can evoke a sense of joy due to the adrenaline rush.

Another segment is securing a livelihood, which, while not dominant, represents a separate motivational group. For them, sports are not just a beloved hobby or activity but also an opportunity to ensure their livelihood and potential financial stability. Finally, recognition was identified as a motivational component. External praises and competition results provide a driving force for winning and personal development. At the professional level, financial recognition plays a role. Lastly, recognition from oneself and teammates is also an important motivational factor.

### 3.2. Gender Differences

Regarding the individual motivational components underlying sport persistence, no significant gender differences were observed (significance: *p* = 0.339; likelihood ratio: 0.171; phi: 0.339; Cramer’s V: 0.339; contingency coefficient: 0.339). However, girls are overrepresented in the motivational component of health maintenance, as they have a higher proportion of this persistence component. Overall, the gender distributions for each persistence component are almost identical (see Table 3).

### 3.3. Importance of the Level of Education

The results of the cross-tabulation analysis show a significant difference in the distribution of the individual factors underlying sport persistence (*p* = 0.046; likelihood ratio: 0.032; phi: 0.069; Cramer’s V: 0.069; contingency coefficient: 0.069). The results are introduced in Table 4. As a prominent factor, the coach is over-represented among young people with an academic education compared to their secondary school peers. In parallel, livelihood and habituation as motivational components are also over-represented among young people in tertiary education.

### 3.4. The Role of Sport Type in the Frequency of Manifestation of Persistence Components

Regarding sport type, there was a trend level difference in the distribution of the overall motivational factors underlying sport persistence (*p* = 0.061; likelihood ratio: 0.015; phi: 0.511; Cramer’s V: 0.511; contingency coefficient: 0.455). Individual athletes were significantly over-represented in those highlighting health maintenance as an underlying motivation, while this motivational factor was under-represented among team athletes. Significant over-representation among individual athletes is observed regarding family motivation, recognition, and self-development. For team athletes, family motivation, coach, livelihood, habituation, sport enthusiasm and competition were slightly over-represented (see Table 5).

### 3.5. Distribution of Sport Level and Persistence Components

Table 6 shows the results concerning the distribution of sport level and persistence components. There is a significant difference in the distribution of the individually defined categories of sport persistence according to the level of sport, i.e., the type of sport as a leisure or competitive activity (*p* = 0.001; likelihood ratio < 0.001; Cramer’s V: 0.001). A strong over-representation of health maintenance and relaxation is observed, which are over-represented among recreational sportspersons. In contrast, sports enthusiasm showed a clear over-representation among competitive sportspersons. In addition, there was a slight over-representation of coach, livelihood, habituation, self-actualization, social relationships, and competition as motivational components towards competitive athletes. Meanwhile, the recognition component was slightly over-represented among recreational athletes.

## 4. Discussion

The research aimed to explore the individual motivational components underlying sport persistence and analyze the trends in these components along with background variables such as gender, level of education, sport type, and sport level. To this end, a semi-structured interview survey was conducted in line, in which the responses of 133 athletes were analyzed along the grounded theory method. The relationship between motivation and persistence in sport participation is multi-faceted, with both internal and external motivational components playing significant roles. The study categorized these motivational factors into two main groups, known as intrinsic motivation (such as health maintenance, habit, self-improvement, and love of sports) and extrinsic motivation (such as family influence, coach, social relationships, competition, and recognition). Sport persistence arises from the interplay between these motivations. Intrinsic factors often provide a more sustainable drive, leading to long-term commitment, while extrinsic motivations, though influential, might lead to less stable participation if the external rewards diminish. Therefore, athletes who combine internal satisfaction with external rewards are likely to maintain their sports involvement over time, demonstrating the essential relationship between motivation and persistence in sports.

The results showed that there were no significant differences in terms of core motivational patterns that promote persistence. Although there may be significant gender differences in the background of sport motivation [53,54], this does not seem to be relevant for persistent sport behavior. Based on the present study, there are no pronounced gender differences in the manifestation of underlying reasons, although previous research suggests that there are typically different reasons behind sport participation. For girls, the most important motivation for participating in physical activity is maintaining and improving health, mobility, and body regeneration. At the same time, boys tend to consider strength, endurance, and body regeneration as the most important motivators for participating in physical activity [55]. According to research by Moradi and colleagues [54], from the perspective of boys, the components of motivation to participate in sports are situational factors, fun, teamwork, fitness, skill development, friendship, energy release, and achievement. Other researchers state that men are often encouraged to be competitive, while women are more directed towards community activities and healthy lifestyles [23]. However, these were not confirmed in our present research concerning sport persistence.

We expected a significant difference in the emergence of individual factors underlying sport persistence based on the participant’s level of education, i.e., whether they had pursued secondary or academic education. However, based on the results of the cross-tabulation analysis, there is only a significant difference in the distribution of individual factors underlying sport persistence. Coaches are over-represented as a salient factor among young people in tertiary education compared to their secondary school peers. Young athletes in tertiary education are often serious about their sporting careers, and the coach’s role may be a key factor [16]. Coaches can help develop talent, prepare for competition, and plan careers, making them of paramount importance to athletes [56,57]. The expertise and support of coaches can be essential in balancing competitive sport and academic commitments. Coaches can not only provide technical and physical training, but also psychological support to athletes. Overall, the findings underscore the role of experienced coaches in fostering talent and providing emotional support, which can be crucial for balancing the demands of both academic and athletic pursuits. Athletes pursuing higher education often face stress and pressure in both sport and academia and coaches can help them with emotional regulation and maintaining motivation. In parallel, livelihood and habituation as motivational components are also over-represented for young people pursuing tertiary education, which is presumably less coded as a function of academic level than as an age-specific characteristic. In Sevil et al.s’ [58] study, high school students showed significantly higher scores on intrinsic motivation than university students and relatively lower scores on amotivation. The time-caused limitations of the students provide less time for leisure activities and sports due to increased academic demands, which suggests that people with intrinsic motivation will be better at time management and will be engaged in sport [30].

There was a trend level difference in the distribution of the general motivational factors underlying sport persistence by sport type. There was a clear overrepresentation of individuals highlighting health maintenance as an underlying motivation among individual athletes, which is less prevalent among team athletes, although some research suggests that team sports may have stronger and additional benefits for mental and social outcomes in adulthood [59,60]. There was a notable overrepresentation among individual athletes for family motivation, recognition, and self-development. For team athletes, family motivation, coach, livelihood, habituation, sportsmanship, and competition were over-represented. The research of Jakobsen and colleagues [61] showed similar results as intrinsic motivation, including the source of pleasure motivation component, was more dominant than team athletes. Moradi and colleagues [54] also showed similar results, adding the motivational dimensions of pleasure source motivation, higher self-improvement, and achievement motivation for individual athletes. Meanwhile, team athletes were more likely to have components of teamwork and belonging to something. Overall, individual athletes were more likely to cite health maintenance as a motivation, possibly reflecting the solitary nature of their training, which allows for personal reflection and self-improvement. Conversely, team athletes often emphasize social relationships and competition, which may stem from the collaborative environment of team sports that fosters a sense of belonging and shared goals. We could confirm Bruner et al.’s [34] and Fraser–Thomas et al.’s [35] results, according to which youth participating in team sports enjoy significant social skills and teamwork benefits, contributing to increased social capital and forming positive social relationships.

A significant difference was found in the distribution of individually defined categories of sport persistence concerning sporting level. Health maintenance and relaxation were found to be over-represented among recreational sportspersons. Recreational athletes are often mindful of the importance of a healthy lifestyle and view sport as part of this lifestyle. As a result, they tend to choose activities that support health and well-being, such as active recreation, movement in nature, and social interaction [62]. However, sport participation also appears significant for recreational athletes, yet it was clearly more relevant among individuals who participated in competitive sport. Competitive athletes often need strong motivation and perseverance to constantly improve and succeed in competitions. Sports enthusiasm and passion for the sport can be driving forces that help them to overcome obstacles and persevere to achieve goals [63]. In addition, coaching, livelihood, habituation, self-actualization, social relationships, and competition as motivational components were also more significant factors among competitive athletes. For competitive athletes, the coach can be of paramount importance in helping them to optimize their performance, develop their technical and tactical skills, and provide preparation and mental support for competitions. With the help of a coach, competitive athletes can be better prepared for competitions and perform better [64,65]. For competitive athletes, sport can, in many cases, be a passion and a livelihood. For professional-level competitors, sports can be the primary source of income; accordingly, it is crucial to succeed and achieve results [66]. In addition, competitive athletes may be more prone to develop a habit of regular training and competitions and make it a permanent part of their lives. This commitment provides them with the opportunity to improve and challenge themselves regularly [67]. Furthermore, competitive athletes often strive to make the most of their sport and achieve the highest levels of success. In this case, self-actualization and personal development can be important motivating factors for them [61]. Competitive athletes are often part of a community or team and develop strong social relationships with their coaches, teammates, and other competitors. These relationships can provide them with support, motivation, and inspiration in sports; the same can be said for competition [68]. Meanwhile, recognition as a motivational component was slightly over-represented among recreational athletes. Recreational athletes often participate in sport because they enjoy the challenges and experiences it provides. They feel a sense of satisfaction through the achievement of personal goals and improvement in their own performance, which can be seen as a form of intrinsic acknowledgment. It can serve as self-validation, where athletes recognize their own progress or effort, reinforcing their self-worth or confidence without relying on feedback from others. In addition, recreational athletes often participate in sport to gain recognition in their interpersonal relationships. Recognition among friends, family members and peers can be significant and motivate them to continue persistent training and activities [69]. Overall, we could see that recreational athletes predominantly cited health maintenance and relaxation as primary motivations, emphasizing the role of sport in promoting a healthy lifestyle and stress relief. This contrasts with competitive athletes, who displayed heightened enthusiasm and a strong desire for self-actualization, indicating a drive for excellence and achievement. This difference may reflect the varying goals and expectations associated with competitive versus recreational sport, where the former often entails a deeper commitment and the pursuit of performance-related outcomes [41,43].

## 5. Conclusions

The current research sought to investigate the motivational factors that contribute to sport persistence in relation to various background variables, including gender, level of education, type of sport, and level of sporting activity. Through the use of semi-structured interviews conducted with 133 athletes and analyzed via the grounded theory approach, the study revealed no notable differences in core motivational patterns for sport persistence across genders, contrary to previous studies that indicated gender-specific reasons for engaging in sports. Variations were observed in relation to educational level, particularly highlighting the significant influence of coaches on athletes pursuing higher education. These individuals, often striving to balance their academic and athletic commitments, derive considerable benefit from the guidance and emotional support provided by their coaches.

Moreover, the type of sport was linked to specific patterns of motivation, whereby individual athletes prioritized health maintenance, familial encouragement, and personal development, in contrast to team athletes who placed greater importance on competition, social connections, and sportsmanship. Competitive athletes exhibited stronger motivation for coach involvement, self-actualization, and habitual training, driven by their passion for sport and the desire for success. Conversely, recreational athletes were more focused on health, relaxation, and intrinsic recognition. This research underscores the intricacies of sport persistence, indicating that forthcoming studies should investigate how these motivational elements change over time and vary across different social and cultural environments. The limitations of the research include sample size and composition, which may limit the generalizability of the results. The sample is not representative, so the results may not be valid in all cases. It should also be noted that interviewees were often unable to distinguish between motivation and persistence, suggesting that further research should be conducted to clarify the conceptual delineation. The qualitative nature of the study, i.e., the research was based on semi-structured interviews due to our aim of discovering the phenomenon of sport persistence and the lack of other assessment tools (i.e., validated questionnaires, accelerometers, parents’ interview, etc.), could limit the results of the study. Although we aimed to recruit a balanced distribution concerning gender, we have to note that males are overrepresented in the sample, serving as a limitation. It is well-known that a balanced representation allows for more robust and generalizable conclusions, particularly in qualitative research where participant experiences and perspectives are central. However, this also reflects on the previous research confirming the higher participation rate of men in sporting activities [70] and men are twice as likely than women to be members of a sport club [16]. Future studies should aim for more balanced gender representation during the sampling process. Further research should also consider the better intersection of gender with other demographic factors (e.g., age, type of sport) to understand how these intersections affect motivational components.

Nevertheless, the research has significant practical relevance, which may affect both the athlete and his/her environment. First of all, coaches and mentors need to understand the individual motivations of athletes and the factors underlying persistence in order to support them more effectively in their development. The results can also help design training methods and motivational strategies. The research shows that athletes’ motivations stem from various factors, including personal development and social experience. Training programs need to take these differences into account to be effective. In addition, the results show that social experience and social recognition are important motivational factors. Therefore, sports clubs and organizations may benefit from organizing programs and events that promote community building and recognition.

## Figures and Tables

**Table 1 jfmk-09-00205-t001:** Composition of the sample by gender, parents’ education level, study level, sport type, and level of sports participation, and their distribution in the sample based on Kolmogorov–Smirnov test.

	N	%	Test Statictic	Significance
Gender			0.393	<0.001
boy	80	60.2		
girl	53	39.8		
Mother’s education			0.307	<0.001
primary	10	7.5		
secondary	71	53.4		
tertiary	52	39.1		
Father’s education			0.342	<0.001
primary	11	8.3		
secondary	81	60.9		
tertiary	41	30.8		
Level of education			0.412	<0.001
secondary	85	63.9		
tertiary	48	36.1		
Type of sport			0.401	<0.001
team sports	82	61.7		
individual sport	51	38.3		
Sporting level			0.439	<0.001
recreational	41	30.8		
competitive	92	69.2		

**Table 2 jfmk-09-00205-t002:** The script of the interview.

Dimension	Variables	Question
Sociodemographic background	age, gender, parents’ education, parents’ labour market status, subjective financial situation, educational level	How old are you? What is your sex? What is the highest educational attainment of your mother/father? How could you evaluate your family’s financial status? Is your mother/father employed? What is the level of your current studies?
Sport-specific questions	sport frequency, sport type, sport level, career stages	How often do you pursue sport? What type of sport do you play most often? Which type of sport do you pursue? At what level do you pursue sport? When did you start to pursue a sport? What kind of stages could you determine in your sports biography?
Competitive activities	activities other than sport (e.g., study, work, other competitions	Besides sport, what kind of other activities could you mention in your life? How can you evaluate your school life and academic achievement? Have you ever been employed/are you employed? Do you pursue any other activities at a high level (e.g., playing a musical instrument)?
Individual motivation	personal drives	Why do you pursue sport? Why are you persistent in pursuing sport? Why do you persist with your sporting activities?
Sport persistence—microsystem	family peers coach	If you consider only your family (excluding other factors), why do you continue to do sports? If you consider only your peers (excluding other factors), why do you continue to do sports? If you consider only your coach (excluding other factors), why do you continue to do sports?
Sport persistence—mesosystem	Family–school relationship Family–sports peers relationship	How does the relationship between your family and your school contribute to your perseverance in sport? How does the relationship between your family and your sporting teammates contribute to your perseverance in sport?
Sport persistence—exosystem	sporting environment	If you only consider your sporting environment, e.g., infrastructure (excluding other factors), why do you continue to do sports
Sport persistence—macrosystem	national culture and traditions	If you only consider national culture and traditions (excluding other factors), why do you continue to do sports?

**Table 3 jfmk-09-00205-t003:** Distribution of individual persistence factors by gender (N = 133).

		Gender		Total
		Boy	Girl	
Family motivation	N	2	0	2
	Row%	100.00%	0.00%	100.00%
	Adjusted residual	1.2	−1.2	
Coach	N	2	0	2
	Row%	100.00%	0.00%	100.00%
	Adjusted residual	1.2	−1.2	
Health promotion	N	9	15	24
	Row%	** 37. ** ** 50% **	** 62. ** ** 50% **	100.00%
	Adjusted residual	−2.5	2.5	
Recognition	N	0	1	1
	Row%	0.00%	100.00%	100.00%
	Adjusted residual	−1.2	1.2	
Becoming a professional athlete	N	1	0	1
	Row%	100.00%	0.00%	100.00%
	Adjusted residual	0.8	−0.8	
Livelihood	N	3	0	3
	Row%	100.00%	0.00%	100.00%
	Adjusted Residual	1.4	−1.4	
Habituation	N	2	1	3
	Row%	66.70%	33.30%	100.00%
	Adjusted residual	0.2	−0.2	
Self-improvement	N	11	7	18
	Row%	61.10%	38.90%	100.00%
	Adjusted residual	0.1	−0.1	
Self-expression	N	1	1	2
	Row%	50.00%	50.00%	100.00%
	Adjusted residual	−0.3	0.3	
Relaxation	N	13	9	22
	Row%	59.10%	40.90%	100.00%
	Adjusted residual	−0.1	0.1	
Loving of sports	N	33	16	49
	Row%	67.30%	32.70%	100.00%
	Adjusted residual	1.3	−1.3	
Social relations	N	2	2	4
	Row%	50.00%	50.00%	100.00%
	Adjusted residual	−0.4	0.4	
Competing	N	1	1	2
	Row%	50.00%	50.00%	100.00%
	Adjusted residual	−0.3	0.3	
Total	Count	80	53	133
	Row%	60.20%	39.80%	100.00%

*Note*: Cells in bold and underlined are under- (adjusted residual < −2) or over-represented (adjusted residual > 2).

**Table 4 jfmk-09-00205-t004:** Distribution of individual persistence factors by level of study (N = 133).

		Level of Education		Total
		Secondary	Tertiary	
Family motivation	N	1	1	2
	Row%	50.00%	50.00%	100.00%
	Adjusted residual	0.4	−0.4	
Coach	N	1	1	2
	Row%	** 50.00% **	50.00%	100.00%
	Adjusted residual	0.4	−0.4	
Health promotion	N	4	20	24
	Row%	** 16.70% **	** 83.30% **	100.00%
	Adjusted residual	−2.2	2.2	
Recognition	N	0	1	1
	Row%	0.00%	100.00%	100.00%
	Adjusted residual	−0.8	0.8	
Becoming a professional athlete	N	0	1	1
	Row%	0.00%	100.00%	100.00%
	Adjusted residual	−0.8	0.8	
Livelihood	N	3	0	3
	Row%	** 100.00% **	** 0.00% **	100.00%
	Adjusted residual	2.3	−2.3	
Habituation	N	3	0	3
	Row%	** 100.00% **	** 0.00% **	100.00%
	Adjusted residual	2.3	−2.3	
Self-improvement	N	7	11	18
	Row%	38.90%	61.10%	100.00%
	Adjusted residual	0.3	−0.3	
Self-expression	N	1	1	2
	Row%	50.00%	50.00%	100.00%
	Adjusted residual	0.4	−0.4	
Relaxation	N	9	13	22
	Row%	40.90%	59.10%	100.00%
	Adjusted residual	0.5	−0.5	
Loving of sports	N	15	34	49
	Row%	30.60%	69.40%	100.00%
	Adjusted residual	−1	1	
Social relations	N	3	1	4
	Row%	75.00%	25.00%	100.00%
	Adjusted residual	1.6	−1.6	
Competing	N	1	1	2
	Row%	50.00%	50.00%	100.00%
	Adjusted residual	0.4	−0.4	
Total	Count	48	85	133
	Row%	36.10%	63.90%	100.00%

*Note*: Cells in bold and underlined are under- (adjusted residual < −2) or over-represented (adjusted residual > 2).

**Table 5 jfmk-09-00205-t005:** Distribution of individual persistence factors by sport type (N = 133).

		Type of Sport		Total
		Team	Individual	
Family motivation	N	2	0	2
	Row%	100.00%	0.00%	100.00%
	Adjusted residual	1.1	−1.1	
Coach	N	2	0	2
	Row%	100.00%	0.00%	100.00%
	Adjusted residual	1.1	−1.1	
Health promotion	N	9	15	24
	Row%	** 37.50% **	** 62.50% **	100.00%
	Adjusted residual	−2.7	2.7	
Recognition	N	0	1	1
	Row%	0.00%	100.00%	100.00%
	Adjusted residual	−1.3	1.3	
Becoming a professional athlete	N	1	0	1
	Row%	100.00%	0.00%	100.00%
	Adjusted residual	0.8	−0.8	
Livelihood	N	3	0	3
	Row%	100.00%	0.00%	100.00%
	Adjusted residual	1.4	−1.4	
Habituation	N	3	0	3
	Row%	100.00%	0.00%	100.00%
	Adjusted residual	1.4	−1.4	
Self-improvement	N	8	10	18
	Row%	44.40%	55.60%	100.00%
	Adjusted residual	−1.6	1.6	
Self-expression	N	1	1	2
	Row%	50.00%	50.00%	100.00%
	Adjusted residual	−0.3	0.3	
Felaxation	N	13	9	22
	Row%	59.10%	40.90%	100.00%
	Adjusted residual	−0.3	0.3	
Loving of sports	N	35	14	49
	Row%	71.40%	28.60%	100.00%
	Adjusted residual	1.8	−1.8	
Social relations	N	3	1	4
	Row%	75.00%	25.00%	100.00%
	Adjusted residual	0.6	−0.6	
Eompeting	N	2	0	2
	Row%	100.00%	0.00%	100.00%
	Adjusted residual	1.1	−1.1	
Total	Count	82	51	133
	Row%	61.70%	38.30%	100.00%

*Note*: Cells in bold and underlined are under- (adjusted residual < −2) or over-represented (adjusted residual > 2).

**Table 6 jfmk-09-00205-t006:** Differences in individual persistence components along the level of sport (N = 133).

		Level of Sport		Total
		Recreational	Competitive	
Family motivation	N	1	1	2
	Row%	50.00%	50.00%	100.00%
	Adjusted residual	0.6	−0.6	
Coach	N	0	2	2
	Row%	0.00%	100.00%	100.00%
	Adjusted residual	−1	1	
Health promotion	N	16	8	24
	Row%	** 66.70% **	** 33.30% **	100.%
	Adjusted residual	4.2	−4.2	
Recognition	N	1	0	1
	Row%	100.00%	0.00%	100.00%
	Adjusted residual	1.5	−1.5	
Becoming a professional athlete	N	0	1	1
	Row%	0.00%	100.00%	100.00%
	Adjusted residual	−0.7	0.7	
Livelihood	N	0	3	3
	Row%	0.00%	100.00%	100.00%
	Adjusted residual	−1.2	1.2	
Habituation	N	0	3	3
	Row%	0.00%	100.00%	100.00%
	Adjusted residual	−1.2	1.2	
Self-improvement	N	5	13	18
	Row%	27.80%	72.20%	100.00%
	Adjusted residual	−0.3	0.3	
Self-expression	N	0	2	2
	Row%	0.00%	100.00%	100.00%
	Adjusted residual	−1	1	
Relaxation	N	11	11	22
	Row%	** 50.00% **	** 50.00% **	100.00%
	Adjusted residual	2.1	−2.1	
Loving of sports	N	7	42	49
	Row%	** 14.30% **	** 85.70% **	100.00%
	Adjusted residual	−3.2	3.2	
Social relations	N	0	4	4
	Row%	0.00%	100.00%	100.00%
	Adjusted residual	−1.4	1.4	
Competing	N	0	2	2
	Row%	0.00%	100.00%	100.00%
	Adjusted residual	−1	1	
Total	Count	41	92	133
	Row%	30.80%	69.20%	100.00%

*Note*: Cells in bold are under- (adjusted residual < −2) or over-represented (adjusted residual > 2).

## Data Availability

Data are available only on request due to ethical restrictions.

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
