# Peer review of "Exploring Individual Components of Sport Persistence in the Light of Gender, Education, and Level and Type of Sport"

_jfmk, 2024, doi:10.3390/jfmk9040205_

Round 1

Reviewer 1 Report

Comments and Suggestions for Authors

This is a very good example of good qualitative research.  It is a large study, 133 high school and college age students in Europe.  The study is an interview process in which themes were detected.  A Cross-tabulation analysis was used to find significance.  I don't know if the study completely found significance, but the findings were important in relation to the difference between recreational, individual, and team sport athletes at different levels of competition.  I enjoyed reading this paper, and believe it is worthy to publish.  

Author Response

Thank you very much for your supportive review. Based on your comments, the following modifications were carried out:

Reviewer 1: This is a very good example of good qualitative research.  It is a large study, 133 high school and college age students in Europe.  The study is an interview process in which themes were detected.  A Cross-tabulation analysis was used to find significance.  I don't know if the study completely found significance, but the findings were important in relation to the difference between recreational, individual, and team sport athletes at different levels of competition.  I enjoyed reading this paper, and believe it is worthy to publish.  

Author: Thank you very much for your kind feedback. Your evaluation is very important for us and we are glad that you enjoyed reading this paper, and believe it is worthy to publish. Thank you very much for your conscientious evaluation!

We hope that we could improve the quality of the manuscript and it can be accepted for publication.

Sincerely,

the Authors

Reviewer 2 Report

Comments and Suggestions for Authors

Review report on the manuscript “An exploratory study of individual motivational components of sport persistence in the light of gender, education and level and type of sport” (jfmk-3204912)

The study addresses an intriguing and relevant topic to the special issue; however, there are significant concerns that need to be addressed to improve the manuscript's overall quality. The following general comments and suggestions are provided to assist in enhancing the paper:

The title is overly long and more importantly does not effectively capture the key aspects of the study. A more concise and representative title is recommended.

The Introduction is overly long and includes unnecessary content, such as information on children (e.g., line 100), as well as some redundant sections. Moreover, key elements are missing, including a clear description of the target population and a review of gaps in the literature.

The paper requires further methodological details. Additional information on the measures, subgroups, ethical considerations, and the study timing should be included. A flowchart or diagram illustrating the study design, classifications, and procedures in the Methods section is highly recommended.

In line 168, a better heading e.g., “Materials and Methods” would be more appropriate.

Section 3.1 appears to belong in the Methods section, rather than the Results. It can help expand the Methods section, which is currently too brief, and enhance the manuscript’s clarity.

Data related to participants' ages are missing in Table 1 and the overall analysis. This information is critical and should be included.

The unbalanced gender distribution among the study groups may have significantly influenced the results. The authors should address this concern with explanations supported by references throughout the manuscript.

It is unclear how confounding and interactive factors were controlled in this study. This should be explicitly addressed in the manuscript.

The manuscript does not indicate whether the questionnaire used in the interview was validated. If the questionnaire was not validated, references for its design should be provided.

The sample size for this type of study is relatively small, which presents a major issue, particularly given the self-reported nature of the data. It should be clearly explained in the sections.

More detailed information is needed regarding supplementary statistical tests (e.g., post-hoc or differentiated tests for subgroup comparisons) and significance levels. Furthermore, a subgroup-based statistical comparison is missing from Table 1. The manuscript lacks adequate quantitative analysis, which is a critical concern, given the nature of the data.

The tables should be redesigned to present the information in a more concise and clear manner.

The Discussion section is well-written, but it does not sufficiently deliver potential explanations or reasoning underlying the study’s findings. This aspect should be a bit expanded.

The manuscript would benefit from the addition of a Conclusion section. Suggestions for future research could also be incorporated here or in the Discussion section.

Regards,

12Sep2024

Comments on the Quality of English Language

The manuscript requires thorough proofreading. In addition, numerous statements, phrases, and terms need to be paraphrased using more advanced scientific grammar, expressions, and terminology to improve readability and professionalism.

Author Response

Thank you very much for your supportive review. Based on your comments, the following modifications were carried out:

Reviewer 2: The study addresses an intriguing and relevant topic to the special issue; however, there are significant concerns that need to be addressed to improve the manuscript's overall quality.

Authors: Thank you very much for your kind feedback. Your suggestions were useful and supported us improving the quality of the paper. Thank you very much for your conscientious evaluation!

Reviewer 2: The title is overly long and more importantly does not effectively capture the key aspects of the study. A more concise and representative title is recommended.

Author: Thank you very much for your kind feedback. We modified the title.

Reviewer 2: The Introduction is overly long and includes unnecessary content, such as information on children (e.g., line 100), as well as some redundant sections. Moreover, key elements are missing, including a clear description of the target population and a review of gaps in the literature.

Author: Thank you very much for your kind feedback. Unfortunately, Reviewer 3 suggested to extend the literature with other relevant theoretical concepts, e.g. extrinsic and intrinsic motivation, definition of sport and physical activity. Therefore, we decided to divide the section to sub-sections which could support the reading of the theoretical background.

Reviewer 2: The paper requires further methodological details. Additional information on the measures, subgroups, ethical considerations, and the study timing should be included. A flowchart or diagram illustrating the study design, classifications, and procedures in the Methods section is highly recommended.

Author: Thank you very much for your kind suggestion. We significantly modified the Methods section and extended its description. The creation and content of the interview script was detailed. Ethical considerations were added, and study design was also added.

Reviewer 2: In line 168, a better heading e.g., “Materials and Methods” would be more appropriate.

Author: Thank you very much for your kind suggestion, we modified the heading.

Reviewer 2: Section 3.1 appears to belong in the Methods section, rather than the Results. It can help expand the Methods section, which is currently too brief, and enhance the manuscript’s clarity.

Author: Thank you very much for the suggestion. However, we modified the Methods section and extended its description. We believe that section 3.1 can be better understood as part of the Results while the content analysis was the first steps of our analysis.

Reviewer 2: Data related to participants' ages are missing in Table 1 and the overall analysis. This information is critical and should be included.

Author: Thank you very much for the suggestion, we completed the manuscript with these data.

Reviewer 2: The unbalanced gender distribution among the study groups may have significantly influenced the results. The authors should address this concern with explanations supported by references throughout the manuscript.

Author: Thank you very much for your kind comment and suggestion. We tried to explain the unbalanced gender distribution and its potential consequences as one of the limitations of the study.

Reviewer 2: It is unclear how confounding and interactive factors were controlled in this study. This should be explicitly addressed in the manuscript.

Author: Author: Thank you very much for the suggestion, we completed the manuscript with these data.

Reviewer 2: The manuscript does not indicate whether the questionnaire used in the interview was validated. If the questionnaire was not validated, references for its design should be provided.

Author: Thank you very much for your kind comment. Since we applied a qualitative research method, we did not use any validated questionnaires. The interview structure and the questions were designed based on our previous analysis of the existing literature focusing on sport persistence. Unfortunately, the cited reference, i.e. our systematic review, is still under review in BMC Psychology, under the third review round.

Reviewer 2: The sample size for this type of study is relatively small, which presents a major issue, particularly given the self-reported nature of the data. It should be clearly explained in the sections.

Author: Thank you very much for your note. However, due to the qualitative nature of the study, we believe that the sample size can be accepted and is typically higher compared to other interview studies. We specified the methodological background of the research in the Methods section.

Reviewer 2: More detailed information is needed regarding supplementary statistical tests (e.g., post-hoc or differentiated tests for subgroup comparisons) and significance levels. Furthermore, a subgroup-based statistical comparison is missing from Table 1. The manuscript lacks adequate quantitative analysis, which is a critical concern, given the nature of the data.

Author: Thank you very much for your kind suggestion. We tried to give more specific information in the regard as well.

Reviewer 2: The tables should be redesigned to present the information in a more concise and clear manner.

Author: Thank you very much for your kind suggestion. We modified the tables.

Reviewer 2: The Discussion section is well-written, but it does not sufficiently deliver potential explanations or reasoning underlying the study’s findings. This aspect should be a bit expanded.

Author: Thank you very much for your kind suggestion. We expended the above-mentioned aspect as well.

Reviewer 2: The manuscript would benefit from the addition of a Conclusion section. Suggestions for future research could also be incorporated here or in the Discussion section.

Author: Thank you very much for your kind suggestion, we added a Conclusions section.

We hope that we could improve the quality of the manuscript and it can be accepted for publication.

Sincerely,

the Authors

Reviewer 3 Report

Comments and Suggestions for Authors

motivational components. It’s a considerable effort from the researchers and the materials involved.

It’s well written and follow the IMRAD design and describes adequately the content of the research protocol. The background of the study is clear and points out the need to further research in this field. The methods and fundings are clear, and well justified as well as the statistical techniques. Moreover, the results and discussion are well written, and present and discuss the main results of the research in adequate manner, clearly presenting the main outcomes of the research, and discussing them accordingly.

After reading I’ d like to suggest some adjustments, some improvements, and corrections.

On line 47: Please, add bibliographic citation for the definition of “sport persistence”.

On line 57-60: Based on research results related to sports habits, it is worth dealing with the issue of persistence in both domestic and international contexts and at various levels of sports. In elite and competitive sports, the role of persistence is unquestionable, as a successful sports career can only be achieved if the athlete remains committed to their activities, ideally with professional training, individually and guided by a coach, and ideally with 61 psychological preparation”. Please, add citations to support these concepts.

On line 73-75: “The role of gender in regular sports activities is complex and depends on various factors, including social, cultural, and biological aspects. Biological differences between genders play a significant role in sports performance, best observed in body composition and hormone levels”.  Please, add citations to support these concepts.

On lines 137 - 142: “The level of sports activity, meaning the frequency, intensity, and duration of participation in sports activities, also plays a significant role in the effectiveness and success of regular sports activities. Beginner-level athletes typically engage in lower-intensity and shorter-duration training sessions, allowing them to gradually adapt to physical activity, avoiding injuries and overtraining. Intermediate-level athletes train more frequently and for more extended periods with higher-intensity training sessions”. Please, add citations to support these concepts.

I would suggest authors to clarify in the Introduction differences between “sports” and “physical activity” to ensure better understanding of the topic.

I would suggest explaining in more detail the procedure used for participants’ recruitment.

Despite the Introduction is well written, I would suggest adding a short paragraph in Methods explaining the reasons why authors used the question blocks reported in the manuscript.

Moreover, since authors mentioned internal and external motivational components in Results, I would strongly suggest adding definitions and references for these components in the Introduction.

On line 415-417: Please, what did authors mean with “intrinsic recognition”? Please, specify it.

In the Discussion, I would suggest authors clarifying the possible relation between motivation and persistence in sport participation. In my opinion, this is one the most important concept of this study.

Since authors used only semi-structured interviews for assessment, I would recommend authors to add this to study limitations. For example, the lack of other assessment tools (i.e., validated questionnaires, accelerometers, parents’ interview, etc.) could limit the results of the study.

I would suggest authors to add Conclusions paragraph (or separate discussion into two paragraph) in which summarize the obtained results and propose methodological implications for coaches, athletes etc.

I would suggest these major changes, and I would like to congratulate the author for the important effort in doing this research and acknowledge the contribution for the understanding of the topic.

Author Response

Thank you very much for your supportive review. Based on your comments, the following modifications were carried out:

Reviewer 3: It’s well written and follow the IMRAD design and describes adequately the content of the research protocol. The background of the study is clear and points out the need to further research in this field. The methods and fundings are clear, and well justified as well as the statistical techniques. Moreover, the results and discussion are well written, and present and discuss the main results of the research in adequate manner, clearly presenting the main outcomes of the research, and discussing them accordingly. After reading I’ d like to suggest some adjustments, some improvements, and corrections.

Authors: Thank you very much for your kind feedback. Your suggestions were useful and supported us improving the quality of the paper. Thank you very much for your conscientious evaluation!

Reviewer 3: On line 47: Please, add bibliographic citation for the definition of “sport persistence”.

Authors: Thank you very much for your kind request. We added citations to support these concepts.

Reviewer 3: On line 57-60: “Based on research results related to sports habits, it is worth dealing with the issue of persistence in both domestic and international contexts and at various levels of sports. In elite and competitive sports, the role of persistence is unquestionable, as a successful sports career can only be achieved if the athlete remains committed to their activities, ideally with professional training, individually and guided by a coach, and ideally with 61 psychological preparation”. Please, add citations to support these concepts.

Authors: Thank you very much for your kind request. We added citations to support these concepts.

Reviewer 3: On line 73-75: “The role of gender in regular sports activities is complex and depends on various factors, including social, cultural, and biological aspects. Biological differences between genders play a significant role in sports performance, best observed in body composition and hormone levels”.  Please, add citations to support these concepts.

Authors: Thank you very much for your kind request. We added citations to support these concepts.

Reviewer 3: On lines 137 - 142: “The level of sports activity, meaning the frequency, intensity, and duration of participation in sports activities, also plays a significant role in the effectiveness and success of regular sports activities. Beginner-level athletes typically engage in lower-intensity and shorter-duration training sessions, allowing them to gradually adapt to physical activity, avoiding injuries and overtraining. Intermediate-level athletes train more frequently and for more extended periods with higher-intensity training sessions”. Please, add citations to support these concepts.

Authors: Thank you very much for your kind request. We added citations to support these concepts.

Reviewer 3: I would suggest authors to clarify in the Introduction differences between “sports” and “physical activity” to ensure better understanding of the topic.

Authors: Thank you very much for your kind suggestion. We introduced the definitions.

Reviewer 3: I would suggest explaining in more detail the procedure used for participants’ recruitment.

Authors: Thank you very much for your kind request. The recruitment procedure was detailed.

Reviewer 3: Despite the Introduction is well written, I would suggest adding a short paragraph in Methods explaining the reasons why authors used the question blocks reported in the manuscript.

Authors: Thank you very much for your kind feedback and suggestion. We modified the Methods section including your kind comment.

Reviewer 3: Moreover, since authors mentioned internal and external motivational components in Results, I would strongly suggest adding definitions and references for these components in the Introduction.

Authors: Authors: Thank you very much for your kind suggestion. We added a paragraph introducing intrinsic and extrinsic motivation.

Reviewer 3: On line 415-417: Please, what did authors mean with “intrinsic recognition”? Please, specify it.

Authors: Thank you very much for your kind notification. Intrinsic recognition was thought to be some kind of self-acknowledgement. We changed and described this term.

Reviewer 3: In the Discussion, I would suggest authors clarifying the possible relation between motivation and persistence in sport participation. In my opinion, this is one the most important concept of this study.

Authors: Authors: Thank you very much for your kind suggestion. We clarified the relation in the Discussion section.

Reviewer 3: Since authors used only semi-structured interviews for assessment, I would recommend authors to add this to study limitations. For example, the lack of other assessment tools (i.e., validated questionnaires, accelerometers, parents’ interview, etc.) could limit the results of the study.

Authors: Thank you very much for your kind suggestion. We added this limitation to the Conclusions section.

Reviewer 3: I would suggest authors to add Conclusions paragraph (or separate discussion into two paragraph) in which summarize the obtained results and propose methodological implications for coaches, athletes etc.

Author: Thank you very much for your kind suggestion, we added a Conclusions section.

Reviewer 3: I would suggest these major changes, and I would like to congratulate the author for the important effort in doing this research and acknowledge the contribution for the understanding of the topic.

Authors: Thank you very much for your kind evaluation once again. We believe your suggestions help us in improving the niveau of the paper.

We hope that we could improve the quality of the manuscript and it can be accepted for publication.

Sincerely,

the Author

Round 2

Reviewer 2 Report

Comments and Suggestions for Authors

While there have been improvements following the revisions from the first-round comments, several critical concerns remain. The manuscript is too lengthy and contains significant structural flaws. A considerable portion of the content should be summarized, and some tables (including the newly added one) should be moved to the supplementary materials. Overall, the paper needs to be more concise to help readers focus on the key aspects.

The introduction is too long and, even longer than the discussion, which is problematic. Approximately 30-40% of the content could either be moved to the discussion or, preferably, summarized.

Newly added statements in lines 234–237 and 213–216 appear to be duplicative. Please refine these sections and carefully check the entire manuscript for any potential duplicates.

Section 2.2 should be divided into multiple subsections to improve clarity and readability.

Section 3.1 is too long and doesn't seem directly relevant to the results. If the authors choose to keep it, it should be significantly condensed.

In response to one of the previous comments, the authors stated: "The interview structure and the questions were designed based on our previous analysis of the existing literature focusing on sport persistence. Unfortunately, the cited reference, i.e. our systematic review, is still under review in BMC Psychology, under the third review round." However, references to relevant "existing literature" are essential and should be included to support the methodology.

Finally, two important comments have not yet been addressed:

1.     1. More detailed information is required regarding supplementary statistical tests (e.g., post-hoc or subgroup-specific tests) and their significance levels. Additionally, subgroup-based statistical comparisons are missing from Table 1. The manuscript lacks sufficient quantitative analysis, which is a major concern given the nature of the data.

2.      2. The tables should be redesigned to present the information more concisely and clearly.

Note: I recommend that the authors specify the modified line numbers in the cover letter when explaining the revisions.

Comments on the Quality of English Language

Moderate editing of English language required. A careful proofreading is necessary. 

Author Response

Thank you very much for your supportive review. Based on your comments, the following modifications were carried out:

Reviewer 2. While there have been improvements following the revisions from the first-round comments, several critical concerns remain. The manuscript is too lengthy and contains significant structural flaws. A considerable portion of the content should be summarized, and some tables (including the newly added one) should be moved to the supplementary materials. Overall, the paper needs to be more concise to help readers focus on the key aspects.

Authors: Thank you very much for your kind feedback. We hope we could make significant modifications that make the paper worth publishing.

Reviewer 2. The introduction is too long and, even longer than the discussion, which is problematic. Approximately 30-40% of the content could either be moved to the discussion or, preferably, summarized.

Authors: Thank you very much for your kind suggestion. We significantly reduced the Introduction section (lines 107-224). We summarised the content of 1.1, 1.2 and 1.3 and integrated the summarised paragraphs to 1. Introduction. The subtitle 1.4 (purpose) was also deleted.

Reviewer 2. Newly added statements in lines 234–237 and 213–216 appear to be duplicative. Please refine these sections and carefully check the entire manuscript for any potential duplicates.

Authors: Thank you very much for your kind comment. We modified the text and deleted duplications (lines 231-234).

Reviewer 2. Section 2.2 should be divided into multiple subsections to improve clarity and readability.

Authors: Thank you very much for your kind suggestion. We divided 2.2. to three subsections (lines 240, 254 and 294).

Reviewer 2. Section 3.1 is too long and doesn't seem directly relevant to the results. If the authors choose to keep it, it should be significantly condensed.

Authors: Thank you very much for your kind suggestion. Section 3.1. is relevant since it introduces the individual persistence factors analysed alongside gender, level of education, level of sporting activity and type of sport. We believe that introducing the content of the dimensions created by content analysis (using GT) supports the understanding of the cross-tabulation analyses.

Reviewer 2. In response to one of the previous comments, the authors stated: "The interview structure and the questions were designed based on our previous analysis of the existing literature focusing on sport persistence. Unfortunately, the cited reference, i.e. our systematic review, is still under review in BMC Psychology, under the third review round." However, references to relevant "existing literature" are essential and should be included to support the methodology.

Authors: Thank you very much for your kind comment. We are glad to inform you that the cited paper has been accepted for publication in BMC Psychology (please find the confirmation of the acceptance attached). For this reason, the paper will be published within a few days. We hope that it can provide a sound basis for the current methodology. However, we also added another qualitative research as a reference (that is also analysed in the given systematic review).

Reviewer 2. More detailed information is required regarding supplementary statistical tests (e.g., post-hoc or subgroup-specific tests) and their significance levels. Additionally, subgroup-based statistical comparisons are missing from Table 1. The manuscript lacks sufficient quantitative analysis, which is a major concern given the nature of the data.

Authors: Thank you very much for your kind notifications. We would like to mention that the qualitative analysis (content analysis) of the research was the most relevant step in this research phase. However, we thought that having between-group comparisons may also support the understanding of the nature of the dimensions. Since the categories and the sociodemographic (gender and level of education) and sport-related variables (type and level of sport) are categorical variables, cross-tabulation analysis were applied. Here, we completed the statistical values (lines 431, 439,  447, 458). In Table1, we introduced the results of the Kolmogorov-Smirnov tests.

Reviewer 2. The tables should be redesigned to present the information more concisely and clearly.

Authors: Thank you very much for your suggestion. However, compared to the previous review round, we deleted the Column% values to make the tables easier to understand. We believe that The number of participants in the cells and Row% showing the percentage within a category as well as adjusted residuals (showing whether relevant  data are present in an under/overrepresented amount) are necessary.

Reviewer 2. Note: I recommend that the authors specify the modified line numbers in the cover letter when explaining the revisions.

Authors: Thank you very much for your kind suggestion, we tried to specify the modified line numbers in our response.

We hope that we could improve the quality of the manuscript and it can be accepted for publication.

Sincerely,

the Author

Round 3

Reviewer 2 Report

Comments and Suggestions for Authors

Thank you for revising!

Comments on the Quality of English Language

Thank you!